# The Invasion of *Galinsoga quadriradiata* into High Elevations Is Shaped by Variation in AMF Communities

**DOI:** 10.3390/plants12183190

**Published:** 2023-09-06

**Authors:** Gang Liu, Ruiling Liu, Benjamin R. Lee, Xingjiang Song, Wengang Zhang, Zhihong Zhu, Yan Shi

**Affiliations:** 1College of Life Sciences, Shaanxi Normal University, Xi’an 710119, China; liuruiling00@163.com (R.L.); songxingjiang521@126.com (X.S.); 18792733016@163.com (W.Z.); zhuzhihong@snnu.edu.cn (Z.Z.); 13838702592@163.com (Y.S.); 2Research Center for UAV Remote Sensing, Shaanxi Normal University, Xi’an 710119, China; 3Changqing Teaching & Research Base of Ecology, Shaanxi Normal University, Xi’an 710119, China; 4Carnegie Museum of Natural History, Pittsburgh, PA 15213, USA; leeb@carnegiemnh.org; 5Department of Biological Sciences, University of Pittsburgh, Pittsburgh, PA 15213, USA; 6Holden Forest and Gardens, Kirtland, OH 44094, USA

**Keywords:** invasive plant, arbuscular mycorrhizal fungi (AMF), plant–AMF interaction, elevational gradients, interspecific competition

## Abstract

Mountain ranges have been previously suggested to act as natural barriers to plant invasion due to extreme environmental conditions. However, how arbuscular mycorrhizal fungi (AMF) affect invasion into these systems has been less explored. Here, we investigated how changes in AMF communities affect the performance of *Galinsoga quadriradiata* in mountain ranges. We performed a greenhouse experiment to study the impact of inoculations of AMF from different elevations on the performance and reproduction of invaders and how competition with native plants changes the effects of invader–AMF interactions. We found strong evidence for a nuanced role of AMF associations in the invasion trajectory of *G. quadriradiata*, with facilitative effects at low elevations and inhibitory effects at high elevations. *Galinsoga quadriradiata* performed best when grown with inoculum collected from the same elevation but performed worst when grown with inoculum collected from beyond its currently invaded range, suggesting that AMF communities can help deter invasion at high elevations. Finally, the invasive plants grown alone experienced negative effects from AMF, while those grown in competition experienced positive effects, regardless of the AMF source. This suggests that *G. quadriradiata* lowers its partnerships with AMF in stressful environments unless native plants are present, in which case it overpowers native plants to obtain AMF support during invasion. Finally, our results indicate that invader–AMF interactions can inhibit invasive range expansion at high elevations, and biotic interactions, in addition to harsh environmental conditions, make high-elevation mountain ranges natural barriers against continued invasion.

## 1. Introduction

The distributions of many invasive plants are shifting to higher elevations within mountain ranges [1,2,3,4,5]. Such dynamics are driven in part by environmental factors such as temperature and precipitation, which change dramatically with elevation and strongly affect plant distributions in mountain environments [6,7]. Many studies have demonstrated that these abiotic environmental factors can affect the invasion trajectory of invasive plant species (e.g., by limiting temperature and nitrogen deposition [8,9]). However, the importance of biotic factors in high-elevation invasion, specifically the importance of interactions with symbiotic fungi, is still far from being fully understood.

Symbiotic fungi, such as arbuscular mycorrhizal fungi (AMF), likely play vital roles in affecting invasion trajectories along elevational gradients [10]. AMF are the predominant type of symbiotic fungi in mountain ecosystems [11] and can form symbiotic relationships with more than 80% of terrestrial plant species [12], including many invasive plant species. They are thought to be a key factor affecting the range expansion of plants in general, especially species that are widespread and minimally inhibited by abiotic conditions [13,14]. Recent studies, for example, suggest that some invasive plants may not be as limited by the drastic climatic gradients found in mountain ranges, perhaps due to high adaptability to temperature and nutrient availability [15,16], as they are limited by interactions with other organisms such as AMF. Furthermore, AMF can promote nutrient uptake and improve the resistance and tolerance of invasive plants to changing environmental factors [17,18,19], potentially lessening the importance of environmental constraints when they do exist. A previous study indicated that AMF could help the invasive species *Solidago canadensis* uptake limited or inaccessible nutrients, such as P [20]. As such, the capacity to establish effective symbiotic relationships with AMF communities is often implicated in the successful invasion of non-native plant species [21,22].

Relationships between invasive plants and AMF communities, however, are also influenced by existing microbial relationships with native plant species [23,24], which also rely on AMF associations to succeed in harsh alpine environments. To invade these regions, invasive plant species often must outcompete native plants for fungal resources. For example, *Vincetoxicum rossicum* was observed to have greater AMF colonization in its invasive range compared to co-occurring native plants, thereby facilitating its widespread invasion [25]. Other studies have shown that AMF can promote the competitive growth ability of invasive species by inhibiting the growth of native species under different soil nutrient levels, especially for congeneric native species [26,27]. Alternatively, some specialist invasive plants rely on the presence of particular AMF species and can be inhibited in their invasion if these species are not present in their invasive range [19,28].

Importantly, many studies investigating invasive plant–AMF interactions do not include plants collected from different populations across heterogeneous environmental gradients [29,30]. Such studies generally ignore that invasion is a dynamic process that occurs across abiotic and biotic gradients [31] and that to succeed, an invader must be able to withstand a wide variety of conditions and biotic interactions. For example, invasive plants will face novel AMF communities as they invade higher elevations, in addition to the novel environmental conditions that are more explored in the invasion literature. It is thus theoretically advantageous for invasive plant species in high-elevation environments to adapt to these novel conditions as they invade [32], but evidence for such a mechanism is thus far limited. It is also possible, however, that invaders are eventually limited by a lack of adequate AMF associations [33], which is most likely to occur at the upper range limits in mountain environments.

In China, the Qinling and Bashan Mountains are important geographic boundaries that act as the ecological transition zone between subtropical and warm temperate zones and are significant for key species of conservation interest (e.g., giant panda) and for maintaining the stability and biodiversity of native ecosystems [34]. Importantly, they are also hypothesized to act as a natural barrier that prevents non-native plant species from moving into northwestern China and Mongolia from the southern and southwestern lowlands [35]. However, the environmental conditions that create these barrier effects (such as cold temperatures and short growing seasons) may be lessened by climate change and anthropogenic disturbances, potentially facilitating invasion into high elevation systems [36]. Nevertheless, it is possible that AMF colonization limitations may continue to prevent invasion into and across this critical habitat.

Here, we investigate the role of biotic interactions with AMF communities and native plant competitors in the invasion trajectory of *Galinsoga quadriradiata* (Asterales: Asteraceae), which is native to tropical America and has been invading China for more than 100 years [36] in high-elevation mountain ranges in central China. Previous studies of ours indicate that the biomass and dispersal traits of this species change along its elevational range [32,37], leading to extensive invasions in the Qinling and Bashan Mountains [9,36]. Most recently, we showed that AMF beta diversity in soils associated with *G. quadriradiata* is strongly affected by the elevation at which the soils were collected in the Qinling and Bashan Mountains [38], supporting that biological interaction strength varies along its invasion trajectory. Nevertheless, an important knowledge gap remains as to how differences in these AMF communities affect the performance of this problematic invader, especially in relation to the presence of native plant species that could compete for AMF associations.

We therefore implemented a greenhouse experiment to investigate how AMF communities impact the growth and performance of *G. quadriradiata* grown from seeds collected along an ~1500-m elevational gradient in the Qinling and Bashan mountain ranges. Specifically, we asked the following questions: (1) How is invasive plant performance affected by inoculation by AMF communities collected from the same and from different elevations? (2) Does AMF inoculant collected beyond the currently invaded range lead to the reduced growth and performance of *G. quadriradiata*? (3) Are the effects of AMF inoculation investigated in the first two questions altered by the additional presence of native plant competitors?

## 2. Results

### 2.1. Biomass Allocation and Mycorrhizal Dependency

The total mass of *G. quadriradiata* was highly dependent on whether native plant species were present or absent. For invaders grown in both mono- and polycultures, total mass was significantly impacted by invader seed source population, AMF inoculation (+/−), and inoculation source (Appendix A). When grown in monoculture and inoculated with AMF from any elevation, the total biomass of *G. quadriradiata* was significantly lower than that in the uninoculated treatment (Figure 1a). Furthermore, there were significant differences among the inoculated treatments, with inoculation from the same elevation resulting in significantly higher biomass than when inoculated with AMF from a different elevation but within the currently invaded range. Inoculation with AMF collected from the high-elevation site (beyond the currently invaded range) resulted in significantly lower biomass than all other treatments.

*Galinsoga quadriradiata* biomass was differently impacted when grown alongside native plant species. Compared to when grown in monoculture, uninoculated invaders grown in polyculture had significantly lower total biomass (Figure 1a). However, total biomass was not significantly different between mono- and polyculture plants in the inoculated treatments (apart from the AMF+S treatment, where biomass was significantly lower when grown in polyculture). 

The seed source population of *G. quadriradiata* (Pop) had a more nuanced effect on total biomass that differed depending on the inoculation treatment (Figure 2a). In general, seeds collected from low elevations tended to produce individuals with higher total biomass than those collected from higher elevations. Mycorrhizal dependency (MD, the net effect of AMF inoculation on the total biomass of the invader) was negative in monoculture, indicating that inoculation had a negative effect on total biomass when the invader was grown alone, but the effect changed to positive when the invader was grown in polyculture (Appendix A).

### 2.2. Root–Shoot Ratio

The invader root–shoot ratio (R:S) was also differently affected depending on whether the invader was grown alone or with native competitors (Figure 1b). In monoculture, there were no differences among any of the inoculation treatments (including the uninoculated control). However, when grown in polyculture with native plants, *G. quadriradiata* allocated significantly more biomass to roots (i.e., higher R:S) when uninoculated compared to all three inoculated treatments. Within the inoculated treatments, significantly more biomass was allocated to roots in the high-elevation inoculation treatment than in the nonself-inoculation treatment (i.e., AMF community from within the currently invaded range, but not from the same elevation from which the seeds were collected).

### 2.3. AMF Colonization Rate

The AMF colonization rate of *G. quadriradiata* grown in monoculture was significantly higher in the three inoculation treatments than in the uninoculated control (Appendix A). However, when grown in polyculture with native plant species, the AMF inoculation from the high-elevation site (beyond the currently invaded elevational range) did not significantly differ from the control. Furthermore, there was a positive relationship between total mass and the AMF colonization rate in both mono- and polycultures (Appendix A). 

Compared to native plants, the total mass of *G. quadriradiata* was greater when inoculated with AMF under each type of culture (Appendix A). The total mass of the three native plants differed significantly by treatment (Appendix A).

### 2.4. Reproduction

The *Galinsoga quadriradiata* reproductive output (seed mass and number of capitula) was also affected by whether plants were grown in mono- or polyculture (Figure 1c,d and Appendix A). In monoculture, seed mass and number of capitula were both significantly greater in the uninoculated control than in all inoculated treatments. Among the inoculated treatments, plants inoculated with AMF collected from the same elevation had a significantly higher reproductive output compared to the other two treatments.

When grown with native competitors, differences in the *G. quadriradiata* reproductive output among the uninoculated control and three inoculation treatments were much lower (Figure 1c,d). Compared to the control, seed mass and number of capitula were each significantly lower for plants inoculated with the high-elevation AMF. However, there was no difference in seed mass between the control and the other two inoculation treatments. There were significantly more capitula for plants grown in the nonself-inoculated treatment compared to the uninoculated control (Figure 1d). Consistent with the trends noted for total biomass, the seed mass, number of capitula, and seed mass ratio all tended to be greater in plants sourced from low-elevation populations than in those from high-elevation populations (Figure 2b–d).

### 2.5. Interspecific Competition Intensity (RII)

The relative interaction index (RII), representing the intensity of competition between *G. quadriradiata* and its native competitors, was significantly affected by AMF inoculation treatment and by elevation of source populations (Table 1). RII was positive under all inoculation treatments (Figure 3), including the uninoculated control, indicating that the overall performance of the invader was negatively impacted by competition with native plant species when grown in polyculture (regardless of AMF community). Furthermore, RII was significantly lower for all AMF inoculation treatments relative to the sterile control (Figure 3), indicating that competitive effects from native plant species most negatively affected invader biomass in the uninoculated control treatment. In soils inoculated with AMF from other elevations, the RII value was significantly lower relative to those in the treatments inoculated with same-elevation soil and those inoculated with high-elevation soil. There were no consistent patterns associated with elevation on invasive RII (Appendix A).

The RII of native plants was higher under all AMF treatments compared to the RII of *G. quadriradiata* (Appendix A), indicating that native species experienced higher competitive effects from the invader than the invader received from them. For the native plants, RII was significantly lower when inoculated with AMF, especially when the soil inoculum was not sourced from the high-elevation site (Appendix A).

### 2.6. Leaf Nutrient Concentration

The leaf nitrogen concentration (LNC) of *G. quadriradiata* differed by both culture and inoculation treatment (Appendix A). LNC was higher in monoculture when inoculated with AMF from within its invaded range (AMF+NS and AMF+S) compared to when uninoculated or when inoculated with AMF from the high-elevation, uninvaded site (AMF+H and AMF−; Figure 4a). However, in polyculture, the LNC from plants inoculated with the high-elevation AMF was significantly lower compared to the LNC in the other three inoculation treatments (which did not differ significantly from each other). The leaf phosphorus concentration (LPC) of *G. quadriradiata* was significantly elevated by AMF inoculation irrespective of whether it was grown in mono- or polyculture (Figure 4b and Appendix A). There were no major differences in LPC among the three inoculation treatments for either culture.

## 3. Discussion

As nonnative plant species move into new environments, the success and extent of their invasion can be limited by environmental conditions and biotic interactions. As far as we know, many current studies focus on the impact of abiotic factors on the dispersal of invasive plants along the altitude, while the impact of biotic factors, such as microorganisms, has not received enough attention. For example, previous studies of plant invasion in high-elevation environments often overlook the importance of mycorrhizal interactions for invasion success (e.g., [8,9]), interactions that have been shown to be critically important for invasion into other environments [19,28]. Here, we conducted a greenhouse study to investigate the importance of interactions with arbuscular mycorrhizal fungi (AMF) for the growth and performance of invasive *Galinsoga quadriradiata* in high-elevation mountain ranges in central China. Specifically, we investigated how AMF inoculants from a range of elevations affect invader performance and whether the presence of native plant competitors alters these dynamics. It is worth noting that we were interested in how changes in the AMF community affect the growth and dispersal of invasive plants in this study; thus, the influence of other types of microorganisms was not considered. Although other types of microorganisms are also important, and the growth and dispersal of invasive plants in nature are the result of the interaction of multiple biological factors, we only focused on the AMF community in this study. Therefore, this article only provides research results on the impact of AMF communities on invasive plants, and perhaps other types of microorganisms can be considered in future studies.

Overall, our results suggest that *G. quadriradiata* is good at adapting to local AMF communities, performing better when grown with inoculum collected from the same elevation site. However, we found that the response to AMF inoculation depended strongly on whether plants were grown alone or with native competitors, with AMF inoculation being more beneficial to the invasive species in competition. 

### 3.1. Invader Adaptation to the AMF Community

Adaptation to novel environmental conditions has been previously documented as a mechanism that facilitates the invasion of nonnative species into mountain ecosystems [37,39,40]. Here, we suggest that this type of adaptation might occur not only in response to novel abiotic conditions but also in response to novel biotic interactions. Our study showed that the biomass accumulation, reproductive allocation, AMF colonization rate, mycorrhizal dependency (MD), and relative interaction index (RII) of the high-elevation populations of invasive *G. quadriradiata* were lower than those of the low-elevation populations when inoculated with AMF (Figure 2, Appendix A and Table 1, Appendix A). This suggests that stressful environmental conditions at high elevations may cause plant–AMF interactions to transition from a net benefit to the plant, to a net negative interaction, where the interaction is no longer helping the plants survive, reproduce, and persist at the population level. This is supported by our previous study, in which we found that the AMF colonization rate of high-elevation populations was lower than that of low-elevation populations in both field surveys and greenhouse experiments [32]. Taken together, these results suggest that *G. quadriradiata* downregulates AMF interactions at high-elevations as a means of maximizing performance in stressful conditions. This reduction in plant–AMF interactions could be either a plastic/behavioral response or a genetically mediated adaptation to stressful environments, similar to other documented trait shifts associated with high-elevation environments [41]. Although we did not quantify genetic differences among the populations of *G. quadriradiata* used in this study, it is important to note that the population-level performance and AMF interaction differences we observed arose from plants that were grown from seeds in the greenhouse, but that never experienced the harsh alpine conditions that their parents did. Thus, we suggest that it is adaptation to both biotic interactions and abiotic environmental drivers that shapes the observed changes in plant-AMF interactions and invasive plant performance across the elevational gradients present in this study, consistent with past research focused on invasive tree species [42]. 

Although the experimental method of AMF extracted is quite mature, we have not determined to what extent the AMF communities we inoculated are representative of natural communities in this study. This may result in our research findings not being directly applicable to interpreting real-life situations in the field. Thus, it is necessary to compare these two AMF communities by the molecular method in the future, and then we can obtain the direct evidence about whether our results obtained are largely representative of natural AMF communities. 

### 3.2. Importance of Elevation for Plant–AMF Interactions

While existing studies have compared changes in plant performance using soil inoculum collected from different populations (e.g., [43]), we are unaware of any studies that have performed this in mountain environments among populations that differed in elevation. We specifically addressed this in our study and found that the elevation from which AMF inoculum was collected had a significant effect on the growth and reproduction of *G. quadriradiata* (Appendix A). Invaders that were inoculated with AMF collected from the same elevation that seeds were collected from had higher total biomass, seed mass, and number of capitula compared to those that were inoculated with AMF from different elevations (Figure 1). 

Furthermore, our results showed that total biomass, seed mass, and number of capitula were significantly lower when *G. quadriradiata* was inoculated with AMF communities from the highest-elevation site, beyond the range of where it was currently invaded (Figure 1). This was even lower than when uninoculated with AMF, suggesting that AMF communities in this region (either species composition or diversity; [42,44]) are at least partially responsible for limiting the invasion of this species.

It is important to note, however, that other studies have found conflicting results. Clavel et al. [44] found that AMF species were consistently present across montane elevational gradients, which would not provide the basis for AMF-limited barriers to invasion at high elevations, as supported in our previous work [32,38]. This could be due to differences between the two studies in elevational range or other factors, such as anthropogenic disturbances (e.g., the highest-elevation site in our study is rarely visited by people and is largely undisturbed). Nevertheless, our findings are supported by other studies that found that the composition and diversity of AMF communities are strongly affected by elevation [42,45], which could facilitate the AMF-mediated limitations to invader success that we posit occurred here. Thus, we conclude that the expansion of invasive *G. quadriradiata* along elevation is limited not only by changing abiotic factors, as is commonly recognized and widely supported [4,46], but also by AMF community dynamics. Although we did not use molecular means to analyze the composition and differences of the AMF communities inoculated at different altitudes in this study, it can be seen from our research results that the AMF inoculated treatment was successful. We think that it is important to consider this part in future study, which can provide more direct evidence for our results.

### 3.3. AMF Community Effects on Nutrient Uptake

AMF inoculation can alter the resource allocation of invasive plants, such as by altering nutrient uptake [20,47]. For example, Qi et al. [20] found that the invasion of *Solidago canadensis* in eastern China was in part facilitated by increased phosphorus uptake associated with AMF colonization. Improved access to limiting resources can thus be an important mechanism by which alien species are able to tolerate stressful novel environments in their invasive range. 

We measured nutrient uptake in *G. quadriradiata* in the forms of leaf nitrogen and leaf phosphorus content (LNC and LPC). When planted in monoculture, LNC was significantly higher when plants were inoculated with AMF communities collected from within the currently invaded range, and it was lower when plants were grown in uninoculated soil or when soil was inoculated with AMF collected from beyond the current observed range (Figure 4). This suggests that *G. quadriradiata* is limited in its capacity to assimilate nitrogen from the soil as mediated through AMF communities.

Interestingly, when *G. quadriradiata* was grown in polyculture with native competitors, there were no significant differences in LNC between uninoculated plants and those that were inoculated with AMF from within the invaded range, suggesting that competition reduces the benefits that *G. quadriradiata* gains from AMF interactions. One possible mechanism that could explain this pattern is if the nitrogen-fixing native plant *Medicago sativa* has a nursery effect on soil nitrogen availability and its utilization by the invader. The mean *G. quadriradiata* LNC when neighboring *M. sativa* was slightly higher than when neighboring non-nitrogen-fixing native plants, but this difference was not statistically significant (Appendix A). This is consistent with previous studies showing that nitrogen-fixing plants can promote nitrogen absorption by their neighbors [48]. Therefore, we concluded that invaders compete for soil nitrogen with native plant communities that they encounter during range expansion.

The LPC of *G. quadriradiata* was significantly higher in inoculated individuals than in uninoculated individuals, regardless of which site the inoculum originated from or which competitor, if any, the invader was grown with (Figure 4 and Appendix A). This indicates that AMF associations consistently promote phosphorus uptake across the currently invaded range. This is consistent with other research that found positive correlations between AMF associations and the phosphorus uptake of invasive plant species [20,49]. 

Taken together, our results suggest that AMF communities extracted from different elevations have consistently positive effects on phosphorus uptake and positive, but inconsistent, effects on nitrogen uptake for *G. quadriradiata* in its invasive range. This suggests, in part, that its invasion into high-elevation environments is more likely to be limited by access to nitrogen than by access to phosphorus. Nitrogen limitations are linked to reduced size, reproduction, and overall performance [50,51,52], which could help explain why the plants inoculated with AMF from the highest elevation site had the lowest biomass and reproduction across all treatments (Figure 1).

### 3.4. Effects of AMF Association Depend on the Presence of Native Plants

Long-term interactions with AMF have been shown to facilitate the invasion of alien plant species across a variety of conditions [21,53]. However, our results showed that AMF affects *G. quadriradiata* growth and reproduction differently depending on whether it is growing in monoculture or in polyculture with native plant competitors. In monoculture, total mass, seed mass, and number of capitula were higher when grown in sterile soil than when inoculated with AMF (regardless of AMF inoculation source). However, in polyculture, the total mass, seed mass, and number of capitula of the invader were higher under the inoculated treatments (Figure 1). As a result, the AMF dependency of *G. quadriradiata* was negative in monoculture but positive in polyculture (Appendix A).

AMF help host plants compete for soil nutrients and water, while as compensation, the host plant provides carbohydrates and lipids for AMF [54,55,56,57]. However, this mutually beneficial interaction can become costly to host plants under certain conditions [58]. When this occurs, plants are theorized to downregulate these interactions [59], thereby eliminating the costs of maintaining the interactions.

Our results suggest that *G. quadriradiata* relies on mycorrhizal associations to successfully invade stressful environments where competition with native plants is present. However, when competition is absent, the invader does not form extensive mycorrhizal associations, implying that it is more beneficial to retain carbon that would otherwise be sent to the AMF. Previous work has shown that AMF associations with invasive plants are partially determined by abiotic factors, e.g., soil phosphorus concentrations [29], but our results go further and suggest that the prevalence of AMF associations and their net effect on invasive plant success are strongly determined by the presence or absence of native competitors.

### 3.5. Effects of Invasion on Native Plant Species

AMF inoculation was more beneficial for invasive plants than for native plants, regardless of the elevation at which the AMF community that plants were inoculated with was collected from (Appendix A). Moreover, the positive effect of AMF on the competitiveness (RII) of the invasive plant was greater than that of native plants (Appendix A). This evidence suggests that part of the reason *G. quadriradiata* has been successful in invading central Chinese mountain ranges is because it can outcompete native plants for AMF associations. 

This is consistent with the enhanced mutualism hypothesis [60], which posits that invasive plants can facilitate range expansion by forming closer symbiotic relationships with AMF than native competitors can. This hypothesis has found support from other studies. For example, in southeast China, AMF inoculation promoted the biomass accumulation of the invasive plants *S. canadensis* and *Triadica sebifera* significantly more than that of native competitors [22,61]. AMF inoculation also promoted the competitiveness of invasive *Echinops sphaerocephalus* in central Europe and reduced the growth advantage and mycorrhizal infection rate of native plants there [62]. Thus, our results further support the enhanced mutualism hypothesis as a mechanism that can enhance invasive species range expansion in general and specifically for the ongoing invasion of *G. quadriradiata* in central Chinese mountain ranges.

## 4. Materials and Methods

### 4.1. Experimental Design

Seeds of five *G. quadriradiata* populations were collected from the Qinling and Bashan Mountains in August 2019 at elevations of 573, 1064, 1526, 1739, and 1930 m.a.s.l. (Figure 5 and Appendix A). For each population, we randomly selected seeds collected from at least 20 mature individuals of *G. quadriradiata* in each population. All seeds were stored in a 4 °C refrigerator until planting in the greenhouse in early May 2020.

Three native species, *Achnatherum splendens* (Poaceae), *Medicago sativa* (Leguminosae), and *Picris hieracioides* (Asteraceae), were chosen to evaluate the effect of competitors on *G. quadriradiata* performance in each soil × population combination. The seeds of *P. hieracioides* were collected from the field, and the seeds of *A. splendens* and *M. sativa* were purchased from sources local to the study location. *M. sativa* is a nitrogen-fixing plant, while *A. splendens* and *P. hieracioides* are not. 

We conducted the experiment in summer 2020 in greenhouses at Shaanxi Normal University, Xi’an, China (E 108.8923°, N 34.1540°). Seeds of invasive and native plants were sown in nursery pots (length 49 cm, width 36 cm, height 3 cm) with sterilized nutrient soil (autoclaved at 0.11 MPa, 121 °C 1 h, twice approximately 3 h apart). In early June 2020, we collected five rhizosphere soil samples from the same sampling points where the seeds were collected, as well as from an additional sixth high-elevation site (at 2391 m.a.s.l.) that was not yet invaded by *G. quadriradiata*. AMF spores from all six locations were extracted from soil samples with tap water using the wet-sieving and decanting method developed by Gerdemann and Nicolson [63]. Specifically, for the soil at each sampling point, first, an appropriate amount of fresh soil was placed in a 1000 mL beaker, tap water was added, and the mixture was stirred well and allowed to stand for approximately 15 s. Then, a sieve with a pore size of 0.5 mm was used to filter the soil suspension to obtain the filtrate, which was then filtered with a sieve with a pore size of 0.038 mm, and this step was repeated until the filtrate was completely filtered. Next, the solution after the second filtration was transferred to a 50 mL centrifuge tube and centrifuged at a speed of 3000 rpm in a conventional centrifuge. After 5 min, the supernatant was removed, an equal amount of 45% sucrose water was added, and the sample was mixed well and centrifuged for 1 min. Finally, the supernatant was filtered with a 0.038 mm sieve to obtain AMF spores, which were transferred to a clean beaker. After that, we checked the extracted AMF spores under the microscope. Healthy spores were stored in different beakers at 4 °C before being used to inoculate soils. Unfortunately, we did not analyze the species composition of the extracted AMF communities by molecular means in this study. However, we do not think this will affect our experimental results, since we are concerned with the effect of the entire extracted AMF community, not a specific strain; thus we just needed to ensure that the AMF community was added to the pot.

After approximately one month of growth (when plants reached approximately 3 cm in height), seedlings were transplanted into plastic pots (height 14 cm, diameter 16 cm) with sterilized soil substrate. The soil substrate was made from a 1:1:1 mixture of sand, nutrient soil, and field soil (Appendix A). Before transplanting, the soil substrate was autoclaved for sterilization (autoclaved twice for one hour at 0.11 MPa and 121 °C). The plastic pots and tools were sterilized by spraying with 0.3% benomyl bactericide.

Two types of culture, mono- and polyculture, were tested (Appendix A). For the monocultures, we planted one individual of *G. quadriradiata* (one of the five populations) or one individual of the native species (one of the three native species) per pot. For the mixed culture, one *G. quadriradiata* (one of the five populations) and one native species (one of the three native species) were planted in the same pot. Seven different AMF treatments were crossed with the culture treatments: (a) 5 mL distilled water with no AMF spores (uninoculated control), (b) inoculated with 5 mL suspension of AMF spores extracted from one of the five sites currently invaded by *G. quadriradiata*, or (c) inoculated with 5 mL suspension of AMF spores extracted from the high-elevation site that was not yet invaded by *G. quadriradiata*. Approximately 200 spores were added to each pot for the AMF treatments. 

For all inoculated pots, we defined those inoculated with AMF spores from the same elevation at which *G. quadriradiata* was collected as self-inoculated (AMF+S) and those inoculated with AMF spores from a different elevation as non-self-inoculated (AMF+NS). Those inoculated with AMF spores from the high-altitude site, which was not yet invaded by *G. quadriradiata*, were separately defined as high-elevation-inoculated (AMF+H). Thus, we had three types of AMF inoculation sources for the invasive plant (AMF+S, AMF+NS, and AMF+H) and two types of AMF inoculation sources for native plants (AMF+ (including all AMF sources except for the high-altitude one) and AMF+H). In total, we had 1210 pots and 1960 individual plants (Appendix A). 

During the experiment, the climate in the greenhouse was characterized by an average relative humidity of 81%, light intensity of 1507 µmol m^−2^ s^−1^ during the daytime (06:00–20:00), and temperature of 26 °C. The times for daylight and night were 14 h and 10 h on average, respectively. Each pot was watered daily with tap water to keep the soil moist.

### 4.2. Data Collection

The greenhouse experiment ended in late September 2020. At the end of the experiment, the number of capitula of each *G. quadriradiata* individual was counted before harvesting. The roots were washed with tap water to remove the soil, and for each treatment, a small amount of fine root was collected from 5 randomly selected individuals of *G. quadriradiata* to measure the AMF colonization rate based on the method described by Liu et al. [32]. Specifically, these 5 cleaned fine roots were stored in formalin-acetic acid-alcohol (FAA) solution and then stored at 4 °C before analysis. Fine root samples were cleared in 20% KOH for 50 min at 60 °C, rinsed with deionized water, and stained with 5% ink–vinegar solution for 30 min at 60 °C following Vierheilig et al. [64]. Then, we randomly selected a subset of 50 root segments that were 1 cm in length from each individual and mounted them on microscope slides. The percentage root length of each subsample colonized by AMF internal hyphae together with vesicles and arbuscules was quantified using the magnified (40 × 10) intersection method based on 75 intersections [65]. In addition, the root, stem, leaves, and seeds of each plant were weighed after the samples were dried to constant weight in an oven at 65 °C for 48 h, and total mass was measured. The root–shoot mass ratio (R:S) was calculated as the ratio of root mass to aboveground mass. The seed mass ratio was calculated as the ratio of seed mass to total mass. The leaf nitrogen concentration (LNC, %) and leaf phosphorus concentration (LPC, %) were analyzed using an autoanalyzer (SEAL AutoAnalyzer III, Norderstedt, Germany). First, 5 replicates were randomly selected from each treatment, ground to powder, and 0.2 g was placed into a digestion tube, and 5 mL of concentrated H_2_SO_4_, 1.85 g of catalyst (Na_2_SO_4_:CuSO_4_ = 10:1), and 3–5 beads of granular zeolite were added. Next, the samples were treated with a graphite digestion instrument (OPSIS), and the specific procedures were: 180 °C for 10 min, 220 °C for 10 min, 350 °C for 10 min, and 420 °C for 60 min. Next, after cooling with the digestion solution, the samples were diluted with deionized water and filtered, the volume was set to 100 mL, and the solution was measured using an autoanalyzer. Finally, the nitrogen and phosphorus contents of these leaf samples were calculated. 

### 4.3. Mycorrhizal Dependency (MD) and Relative Interaction Index (RII)

The mycorrhizal dependency (MD) of *G. quadriradiata* was calculated according to the following equation [66]:
(1)$$\mathrm{MD}=\frac{{(M}_{\mathrm{AMF}+})-(M_{\mathrm{AMF}-})}{M_{\mathrm{AMF}-}}$$
where M_AMF+_ is the total mass of *G. quadriradiata* inoculated with AMF spores, and M_AMF-_ is the total mass of uninoculated *G. quadriradiata*. MD represents the intensity and direction of the interaction between *G. quadriradiata* and AMF. MD < 0 indicates a negative association between AMF colonization and total biomass, whereas MD > 0 indicates that the focal individual’s biomass is positively associated with AMF colonization.

We also defined the relative interaction index (RII) between *G. quadriradiata* and the native competitors according to the following equation:
(2)$$\mathrm{RII}=\frac{M_{\mathrm{Single}}-M_{\mathrm{Mixed}}}{M_{\mathrm{Single}}+M_{\mathrm{Mixed}}}$$
where M_Single_ is the total mass of *G. quadriradiata* grown in monoculture and M_Mixed_ is the total mass of *G. quadriradiata* grown in polyculture. Therefore, RII represents the intensity of competition between the invasive *G. quadriradiata* and its native competitors. If RII < 0, *G. quadriradiata* experiences a positive impact (i.e., facilitation) from the native plant species; if RII > 0, *G. quadriradiata* instead experiences a negative impact (i.e., competition).

### 4.4. Statistical Analysis

Generalized linear mixed-effects models (GLMMs) were used to test the effects of population (Pop), AMF treatment, culture (single or mixed), and native competitors (N-fixing or non-N-fixing) on the AMF colonization rate, MD, growth performance (total mass, R:S, seed mass, seed mass ratio, number of capitula, LNC, and LPC), and interaction strength (RII) between *G. quadriradiata* and its native competitors.

For the majority of response variables (total mass, R:S, seed mass, seed mass ratio, number of capitula, LNC, LPC, and AMF colonization rate), Pop (5 populations of invasive plant), Culture (single or mixed), Competitor types (nitrogen-fixing or not, nested in Culture), AMF treatments (AMF+ or AMF−), and inoculation source (S, NS or H; nested in the AMF+ treatment) were assigned as fixed factors. Pop, Culture, Competitor type (nested in Culture), AMF treatments, and inoculation source (nested in AMF treatments) were fixed factors in the RII model. Pop, Culture, Competitor types (nested in Culture), and inoculation source were fixed factors in the MD model. For all groups, Competitor species was taken as a random factor nested in Culture. We performed all analyses using SAS 9.3 (SAS Institute Inc., Cary, NC, USA) and R 4.0.3 [67].

## 5. Conclusions

*Galinsoga quadriradiata* is a problematic invasive species in China, and whether this species will be able to invade past high mountain ranges into critical conservation areas in central Asia is an ongoing question. Here, we demonstrated that plant–AMF interactions change along elevational invasion trajectories, suggesting that, although AMF interactions are likely to have facilitated *G. quadriradiata* invasion at low altitudes, it is unlikely that they will continue to facilitate invasion past its currently invaded range. Specifically, AMF from low-elevation communities were found to improve *G. quadriradiata* growth and reproduction, but AMF from high-elevation communities that currently lie outside of the invaded range negatively affected these same performance metrics, especially when *G. quadriradiata* was grown in the absence of native competitors. 

Furthermore, *G. quadriradiata* performed best when inoculated with AMF collected from the same elevation where they were collected, suggesting population-level adaptation to mycorrhizal communities. In general, compared to native competitors, the invasive plant was more positively affected by AMF symbiosis, especially when grown in polyculture, suggesting that invasion has been largely facilitated by a superior capacity to compete for AMF associations. AMF associations play a highly nuanced role in the range expansion of invasive plants into high-elevation environments: the same associations that facilitate invasion at low elevations can deter invasion in montane environments. Importantly, this supports the theory that high-elevation mountain ranges, such as the Qinling and Bashan Mountains in central China, can be valuable natural barriers to plant invasion. Since the results of greenhouse experiments are difficult to extrapolate to natural habitats, future research should focus on how the interaction between AMF communities and invasive species occurs in natural habitats and how other biotic and abiotic factors affect the interaction between them, which would provide a theoretical basis for the prevention and control of invasive species.

## Figures and Tables

**Figure 1 plants-12-03190-f001:**
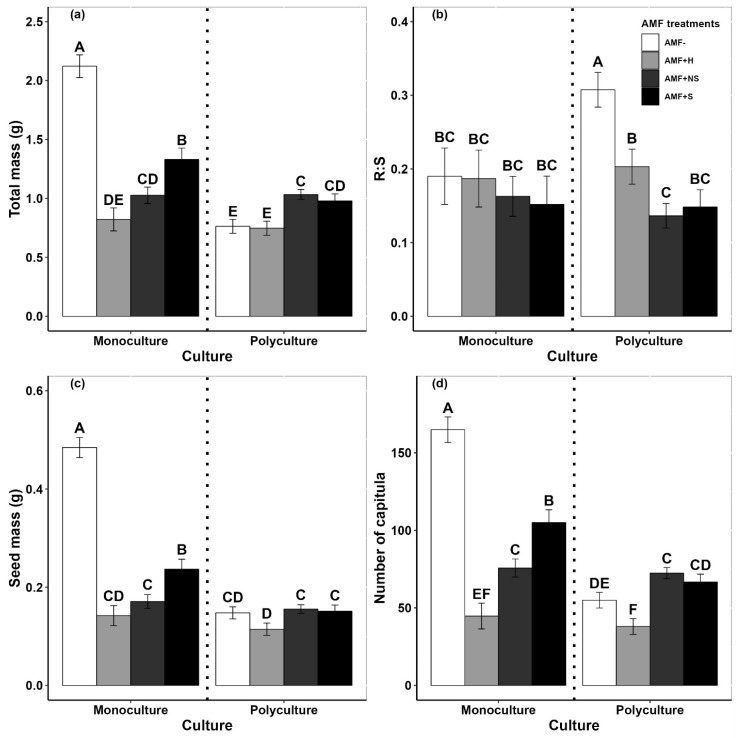
The effects of different treatments on *Galinsoga quadriradiata*: (**a**) total biomass, (**b**) R:S (root-shoot ratio), (**c**) seed mass, and (**d**) number of capitula. All panels are separated by culture (grown alone or with native competitors). AMF inoculation treatments included uninoculated (AMF−) or were inoculated with AMF from the same elevation (AMF+S), a different elevation but within the currently invaded range (AMF+NS), or from the high-elevation, uninvaded site (AMF+H). Letters indicate significant differences between Culture × AMF treatment combinations within each panel (*p* < 0.05), and whiskers indicate standard error around mean values.

**Figure 2 plants-12-03190-f002:**
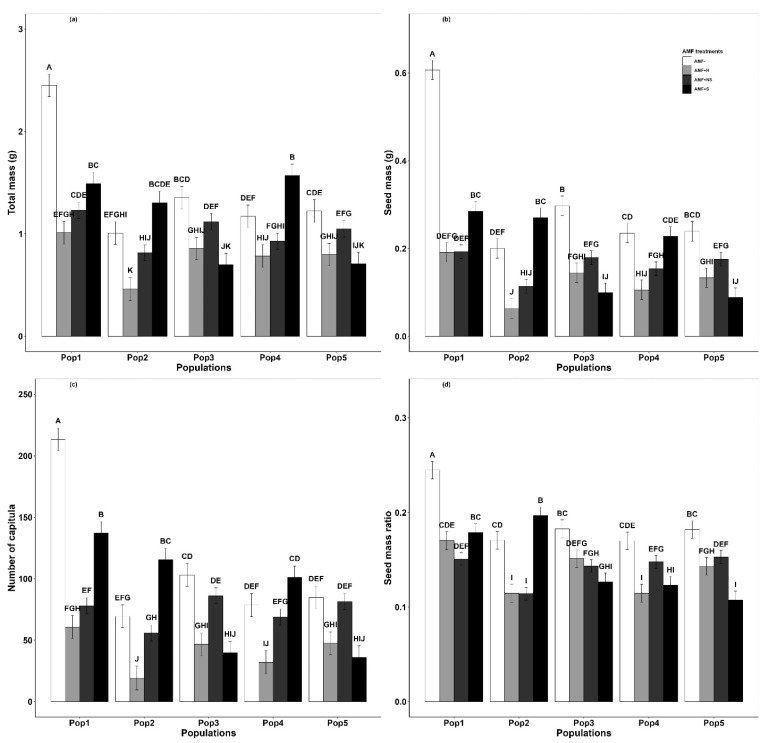
Effects of AMF inoculum treatment on *Galinsoga quadriradiata*: (**a**) total biomass, (**b**) seed mass, (**c**) number of capitula, and (**d**) seed mass ratio, with effects separated by the different populations that seeds were collected from (as shown in Figure 5). Populations ranged in elevation from 573 m.a.s.l. (Pop1) to 1930 m.a.s.l. (Pop5), with elevation increasing with population identification number. AMF inoculation treatments, significance labels, and error bars are the same as described for Figure 1.

**Figure 3 plants-12-03190-f003:**
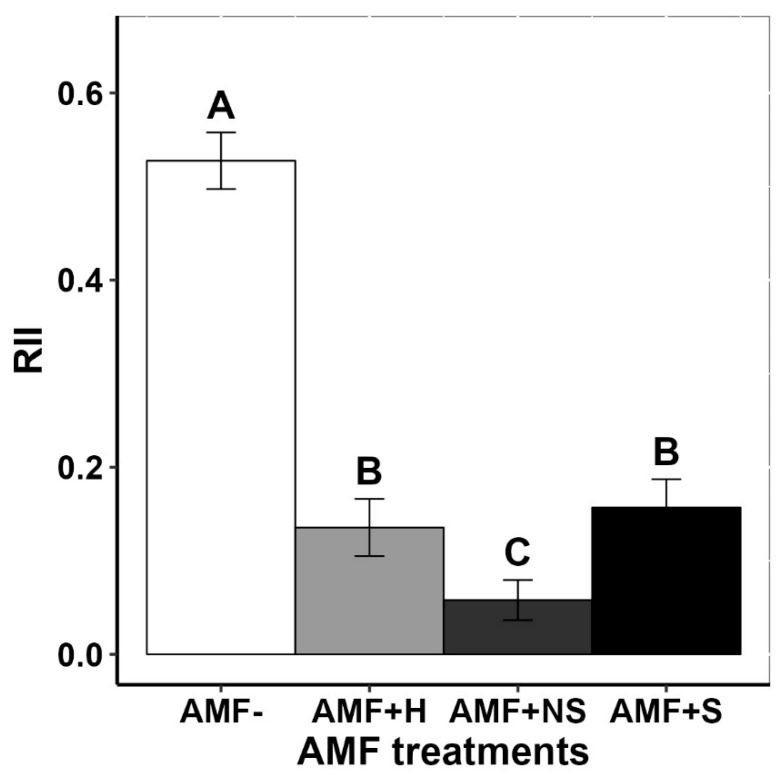
The effect of different AMF treatments on the RII (relative interaction index) of *Galinsoga quadriradiata* populations grown in polyculture with native competitors. AMF inoculation treatments, significance labels, and error bars are the same as described for Figure 1.

**Figure 4 plants-12-03190-f004:**
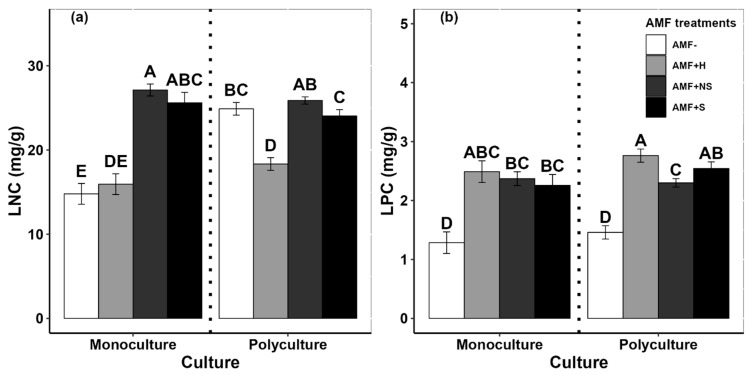
Effects of AMF inoculation treatments on the (**a**) leaf nitrogen concentration (LNC) and (**b**) leaf phosphorus concentration (LPC) of *Galinsoga quadriradiata* grown in mono- or polyculture. AMF inoculation treatments, significance labels, and error bars are the same as described for Figure 1.

**Figure 5 plants-12-03190-f005:**
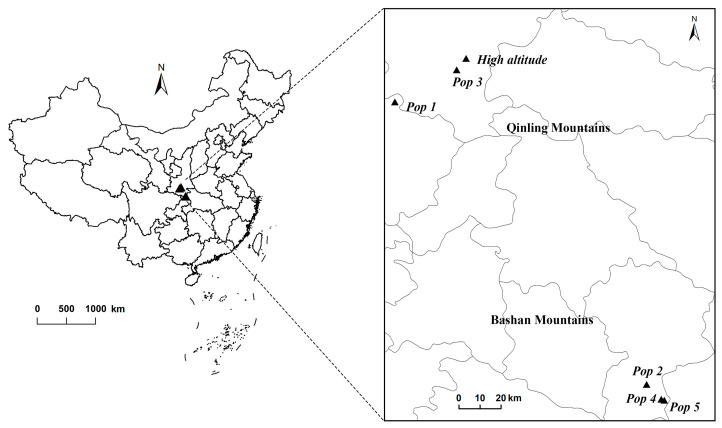
The distribution of sample collection locations for rhizosphere soil and *Galinsoga quadriradiata* seed in the Qinling and Bashan mountain ranges in central China.

**Table 1 plants-12-03190-t001:** Results of generalized linear mixed models (GLMMs) on the RII (relative interaction index) of *Galinsoga quadriradiata*. Pop: population of *G. quadriradiata*. AMF: inoculated (AMF+) and uninoculated (AMF−). Inoculation source (InS): (S), inoculated with AMF spores from the same elevational population of *G. quadriradiata*; (NS), inoculated with AMF spores from a different elevational population of *G. quadriradiata*; (H), inoculated with AMF spores from the high-altitude site that has not yet been invaded by *G. quadriradiata*. (Cul): mono- or polyculture. (Competitors): the three native plants, *Achnatherum splendens*, *Medicago sativa*, and *Picris hieracioides*. (CompTy): nitrogen-fixing or non-nitrogen-fixing native competitor. Fixed factors: Pop, CompTy nested in Cul, AMF, and InS nested in AMF; random factor: Competitors nested in Cul. Effects were considered significant at *p* < 0.05, indicated by bold font.

Effect		Pop	CompTy(Cul)	Pop × CompTy(Cul)	AMF	Pop × AMF	CompTy × AMF(Cul)	Pop × CompTy × AMF(Cul)	InS(AMF)	Pop × InS(AMF)	CompTy × InS(Cul × AMF)	Pop × CompTy × InS(Cul × AMF)
RII	*df*	4702	1702	4702	1702	4702	1702	4702	2702	8702	2702	8702
	*F*	18.56	0.03	2.34	171.13	7.48	3.44	1.81	5.27	6.52	2.55	2.46
	*p*	**<0.0001**	0.861	0.054	**<0.0001**	**<0.0001**	0.064	0.125	**0.005**	**<0.0001**	0.079	**0.012**

## Data Availability

The data that support the findings of this study are available from the corresponding author upon reasonable request. If the paper is accepted for publication, the data will be permanently archived at: https://www.scidb.cn/en (accessed on 1 October 2022).

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
