# Peer review of "The Invasion of Galinsoga quadriradiata into High Elevations Is Shaped by Variation in AMF Communities"

_plants, 2023, doi:10.3390/plants12183190_

Round 1

Reviewer 1 Report

In this study, a greenhouse experiment has been conducted to investigate how AMF communities impact the growth and performance of G. quadriradiata, with focus on the elevation gradient of Qinling and Bashan Mountains.

Overall, the paper is well written. But the major concern is that the conclusion drawn herewith for differential response to particular AMF need to be supported by some additional information / clarification.

There is no information if the AMF at different elevations were same or different. Was there any attempt to identify these AMF?

There are reports that the different AMF recruit different microbiome in rhizosphere, and hence the effect may be not due to AMF ‘only’ (https://www.nature.com/articles/s41598-017-14487-6)

All the pots were kept at same green house conditions. As this study has ‘elevation’ as one of the variables, its important to address that different elevation will have different temperatures and soil conditions. How does keeping all pots at same conditions justifies the gradient of elevation.

Which soil was used for pot trials? This need to be mentioned. Was is also applied according to the gradient? Were the physico-chemical properties of soil varied with the elevation?

Line 414: “Two types of culture, mono- and polyculture, were tested (Figure S1). For the mono cultures, we planted one individual of G. quadriradiata (one of the five populations) or one individual of native species (one of the three native species) per pot. For the mixed culture, one G. quadriradiata (one of the five populations) and one native species (one of the three native species) were planted in the same pot.”  ……This is not clear. What is ‘population’ being referred here with, (also in table 1) ? Also, its not clear with this if the total number of plants in each pot were different?

 The use of term ‘strongly’ may be re-considered, in the title.

need to use appropriate terms for clarity. 

Author Response

#Reviewer 1

Comments and Suggestions for Authors

In this study, a greenhouse experiment has been conducted to investigate how AMF communities impact the growth and performance of G. quadriradiata, with focus on the elevation gradient of Qinling and Bashan Mountains.

Overall, the paper is well written. But the major concern is that the conclusion drawn herewith for differential response to particular AMF need to be supported by some additional information / clarification.

Response: Thank you for your suggestions, we believe that this conclusion is based on our experimental design, but we understand that such a conclusion needs to be supported by more direct information, and this has become a question for our future experiments to explore, that is, to explain this phenomenon by mechanism.

There is no information if the AMF at different elevations were same or different. Was there any attempt to identify these AMF?

Response: Thank you for your suggestions, our previous study has indicated that the rhizosphere and root AMF community of this invasive species were changed along elevation, please check the introduction section of Line 92 – 94.

There are reports that the different AMF recruit different microbiome in rhizosphere, and hence the effect may be not due to AMF ‘only’ (https://www.nature.com/articles/s41598-017-14487-6)

Response: As you mentioned, it is true that AMF also interacts with other microbial communities, but in this paper, we focus on the effects of AMF, so in this study, we used greenhouse experiments to control the variables of AMF. The question you mentioned should be a study that considers different microbial types.

All the pots were kept at same greenhouse conditions. As this study has ‘elevation’ as one of the variables, its important to address that different elevation will have different temperatures and soil conditions. How does keeping all pots at same conditions justifies the gradient of elevation.

Response: Perhaps you didn't really understand our experimental design, as you stated, different altitudes have different temperatures and soil conditions. In addition, biological factors such as microbial composition also vary greatly, however it is difficult to account for all these variables in a single study. In our study, we first considered the influence of biological factors on the altitude gradient, that is, the effect of AMF community changes on invasive plants (AMF community changes as independent variables). The second consideration is the combined response of the invasive species to abiotic and biotic factors along the altitude gradient (populations from different altitude gradients). Therefore, we believe that all pots should be placed under the same conditions to explore how AMF communities at different altitudes affect the growth of invasive plants, and how populations at different altitudes respond to these changes.

Which soil was used for pot trials? This need to be mentioned. Was is also applied according to the gradient? Were the physico-chemical properties of soil varied with the elevation?

Response: thank you for your suggestion, the physico-chemical properties of soil were the same in all pots, we had mentioned in the Table S2, please check the Materials and Methods section of Line 350 – 351 and the supplement part of Table S2.

Line 414: “Two types of culture, mono- and polyculture, were tested (Figure S1). For the mono cultures, we planted one individual of G. quadriradiata (one of the five populations) or one individual of native species (one of the three native species) per pot. For the mixed culture, one G. quadriradiata (one of the five populations) and one native species (one of the three native species) were planted in the same pot.” ……This is not clear. What is ‘population’ being referred here with, (also in table 1) ? Also, its not clear with this if the total number of plants in each pot were different?

Response: Population refers to different populations of the same species. According to previous studies, with the increase of altitude, the traits of populations of the same species at different altitudes will change to adapt to the changes of biological and abiotic factors, such as the decrease of height and the increase of leaf thickness. In this study, we selected 5 populations with different altitudes, and the seeds of these 5 populations were collected at 5 altitudes. In addition, our experimental design included two planting methods to explore the effect of the presence of native plants on the interaction of invasive plants with AMF. Single culture refers to a single plant in a pot, including five populations of invasive plants and three different native species. Mixed culture involves planting two plants in a pot, one of which is an invasive species from one of five populations, one is a native plant, one of three different native plants.

The use of term ‘strongly’ may be re-considered, in the title.

Response: We have deleted ‘strongly’, please check the title of Line 1.

Reviewer 2 Report

Interesting manuscript, well written, some errors are listed below:

32 there is: …Taken together, our results indicate that invader–AMF; should be: …Finally, our results indicate that invader–AMF

78  there is: … a dynamic process that…; should be: … a dynamic process which…

Author Response

#Reviewer 2

Comments and Suggestions for Authors

Interesting manuscript, well written, some errors are listed below:

32 there is: …Taken together, our results indicate that invader–AMF; should be: …Finally, our results indicate that invader–AMF

Response: thank you for your suggestion, we have modified this sentence, please check the abstract section of Line 35.

78 there is: … a dynamic process that…; should be: … a dynamic process which…

Response: thank you for your suggestion, we have modified this sentence, please check the introduction section of Line 72.

Round 2

Reviewer 1 Report

I could see the authors have responded to the concerns. However, the responses are directed to reviewers as explanations, but does not reflect in the revised manuscript. Its advised that the authors should give due considerations to the concerns as raised and provide justified descriptions in the revision. 

1. line 349-359: refer to the method of AMF extraction and application. The point raised was for the AMF used in this study. AMF isolation and application here, not characterization. So a justification need to be given in the manuscript (Not about the AMF diversity study reported previously as given in introduction).  

2. The response is unsatisfactory, and even so, it should be addressed in manuscript and not to reviewer only. The role of other microbes (other than AMF) had been ignored in this study. This should be presented with suitable justification, may be in discussion.  

3. The experimental design is well understood and that is why its important to address this, as this justification is not clear from descriptions given in the manuscript. As explained - "The second consideration is the combined response of the invasive species to abiotic and biotic factors along the altitude gradient (populations from different altitude gradients). Therefore, we believe that all pots should be placed under the same conditions to explore how AMF communities at different altitudes affect the growth of invasive plants, and how populations at different altitudes respond to these changes." This may not be a logical explanation to some of the readers as abiotic factors varies with the altitude, but as the authors asserts that this is their belief, so this justification is required to be mentioned in the manuscript in discussion section. 

4. The authors should understand that in table S1, five different soil type has been mentioned as per different altitude, while from table S2, or anywhere else in the manuscript, its not given which soil was used for the purpose of planting. That is why the point was raised, and should be addressed, in the revised manuscript, as its apparent that there were different soil samples as per difference in altitude.  

Author Response

Comments and Suggestions for Authors

I could see the authors have responded to the concerns. However, the responses are directed to reviewers as explanations, but does not reflect in the revised manuscript. Its advised that the authors should give due considerations to the concerns as raised and provide justified descriptions in the revision.

Response: Thank you for your suggestion. We have modified some sentences, please check the whole text.

  1. line 349-359: refer to the method of AMF extraction and application. The point raised was for the AMF used in this study. AMF isolation and application here, not characterization. So a justification need to be given in the manuscript (Not about the AMF diversity study reported previously as given in introduction).

Response: Thank you for your suggestion. We have added some sentences to explain that we did not quantify the characteristics of the AMF community. Please check the Lines 371 – 374.

  1. The response is unsatisfactory, and even so, it should be addressed in manuscript and not to reviewer only. The role of other microbes (other than AMF) had been ignored in this study. This should be presented with suitable justification, may be in discussion.

Response: Thank you for your suggestion. In our study, we only focused on the AMF community of microorganisms. While we don't understand why you keep insisting on the influence of other microorganisms, we have added some sentences to point out that we did not consider the influence of other microorganisms. Please check the Lines 199 – 205.

  1. The experimental design is well understood and that is why its important to address this, as this justification is not clear from descriptions given in the manuscript. As explained - "The second consideration is the combined response of the invasive species to abiotic and biotic factors along the altitude gradient (populations from different altitude gradients). Therefore, we believe that all pots should be placed under the same conditions to explore how AMF communities at different altitudes affect the growth of invasive plants, and how populations at different altitudes respond to these changes." This may not be a logical explanation to some of the readers as abiotic factors varies with the altitude, but as the authors asserts that this is their belief, so this justification is required to be mentioned in the manuscript in discussion section.

Response: Thank you for your suggestion. We believe our logic will be better understood when read in conjunction with the Introduction section, and we have added some sentences in the discussion explaining why we focus on biological factors. Please check the Lines 191 – 193.

  1. The authors should understand that in table S1, five different soil type has been mentioned as per different altitude, while from table S2, or anywhere else in the manuscript, its not given which soil was used for the purpose of planting. That is why the point was raised, and should be addressed, in the revised manuscript, as its apparent that there were different soil samples as per difference in altitude.

Response: Thank you for your suggestion. Six different soil type had been only used for extracted AMF community from in situ soil, which we have described in Lines 357 – 361. The information of table S1 only about the sample information of this study. In addition, the plant soil substrate was from a 1:1:1 mixture of sand, nutrient soil, and field soil. The sand and nutrient soil are purchased from the company, and the field soil comes from the open space around the greenhouse. We have described it at Lines 376 – 378.

Round 3

Reviewer 1 Report

I understand that the authors are skeptical about role of other organisms, because it has not been studied. But that exactly is the reason it was suggested to mention. The invasive species are known to alter environmental microbiomes. This altered microbiomes (including culturable and unculturable) has effect on plant adaptability, growth and survival. So the effects being claimed solely because of AMF is not a complete information. This perspective may be, part of future studies by the authors. Some interesting papers on this are given that may be useful to authors. 

1. Torres, N., Herrera, I., Fajardo, L. et al. Meta-analysis of the impact of plant invasions on soil microbial communities. BMC Ecol Evo 21, 172 (2021). https://doi.org/10.1186/s12862-021-01899-2

2. Malacrinò A, Sadowski VA, Martin TK, Cavichiolli de Oliveira N, Brackett IJ, Feller JD, Harris KJ, Combita Heredia O, Vescio R, Bennett AE. Biological invasions alter environmental microbiomes: A meta-analysis. PLoS One. 2020 Oct 22;15(10):e0240996. doi: 10.1371/journal.pone.0240996. 

3. Sokornova, S.; Malygin, D.; Terentev, A.; Dolzhenko, V. Arbuscular Mycorrhiza Symbiosis as a Factor of Asteraceae Species Invasion. Agronomy 202212, 3214. https://doi.org/10.3390/agronomy12123214

4. Rout ME, Callaway RM. Interactions between exotic invasive plants and soil microbes in the rhizosphere suggest that 'everything is not everywhere'. Ann Bot. 2012 Jul;110(2):213-22. doi: 10.1093/aob/mcs061. 

The authors have addressed the issues in the manuscript. It may be accepted.